# Clinical usefulness of brief screening tool for activating weight management discussions in primary cARE (AWARE): A nationwide mixed methods pilot study

**Evan Atlantis**[1,2]*, **James Rufus John**[3,4,5], **Paul Patrick Fahey**[1], **Samantha Hocking**[6,7], **Kath Peters**[8]

1 School of Health Sciences, Western Sydney University, Penrith, New South Wales, Australia, 2 Discipline of Medicine, Nepean Clinical School, Faculty of Medicine and Health, The University of Sydney, Nepean, New South Wales, Australia, 3 BestSTART-SWS, South Western Sydney Local Health District, Liverpool, New South Wales, Australia, 4 Academic Unit of Child Psychiatry, School of Psychiatry, Faculty of Medicine, University of New South Wales, Kensington, New South Wales, Australia, 5 Ingham Institute of Applied Medical Research, Liverpool, New South Wales, Australia, 6 The Boden Collaboration for Obesity, Nutrition, Exercise & Eating Disorders, Charles Perkins Centre, The University of Sydney, Camperdown, New South Wales, Australia, 7 Metabolism & Obesity Services, Royal Prince Alfred Hospital, Camperdown, New South Wales, Australia, 8 School of Nursing and Midwifery, Western Sydney University, Penrith, New South Wales, Australia

* E.Atlantis@westernsydney.edu.au

**Data Availability Statement:** All relevant data are within the manuscript and its Supporting Information files.

## Abstract

### Objective

The Edmonton Obesity Staging System (EOSS) is based on weight related health complications among individuals with overweight and obesity requiring clinical intervention. We aimed to assess the clinical usefulness of a new screening tool based on the EOSS for activating weight management discussions in general practice.

### Methods

We enrolled five General Practitioners (GPs) and 25 of their patients located nationwide in metropolitan areas of Australia to test the feasibility, acceptability, and accuracy of the new 'EOSS-2 Risk Tool', using cross-sectional and qualitative study designs. Diagnostic accuracy of the tool for the presence of EOSS ≥2 criteria was based on clinical information collected prospectively. To assess feasibility and applicability, we explored the views of GP and patient participants by thematic analysis of transcribed verbatim and de-identified data collected by semi-structured telephone interviews.

### Results

Nineteen (76%) patients were aged ≥45 years, five (20%) were male, and 20 (80%) were classified with obesity. All 25 patients screened positive for EOSS ≥2 criteria by the tool. Interviews with patients continued until data saturation was reached resulting in a total of 23 interviews. Our thematic analysis revealed five themes: *GP recognition of obesity as a*

**Funding:** This pilot work was supported by grants from iNova Pharmaceuticals (Australia) Pty Ltd (https://inovapharma.com/), in partnership with the National Association of Clinical Obesity Services Incorporated (https://www.nacos.org.au/) and Western Sydney University (https://www. westernsydney.edu.au/) (P00026836: EA, JRJ, PPF, and KP). The funders had no role in study design, data collection and analysis, decision to publish, or preparation of the manuscript.

**Competing interests:** EA was the Founding President, and now serves as the Secretary, of NACOS. He has received honoraria from Novo Nordisk for speaking and participating at meetings. He has received unrestricted research funding from Novo Nordisk and iNova on behalf of NACOS. JRJ received payment for his role as the project manager through casual employment contracts at Western Sydney University. SH has received honoraria from Novo Nordisk, iNova, Sanofi, Lilly, Boehringer Ingelheim, Servier, MSD and Astra Zeneca for seminar presentations. She has served on advisory boards for Lilly, iNova, Pfizer and Novo Nordisk. She has received research funding from Novo Nordisk. She is the current the President of NACOS. PPF and KP declared that no competing interest exists.

*health priority* (GPs expressed strong interest in and understanding of its importance as a health priority); *obesity stigma* (GPs reported the tool helped them initiate health based and non-judgmental conversations with their patients); *patient health literacy* (GPs and patients reported increased awareness and understanding of weight related health risks), *patient motivation for self-management* (GPs and patients reported the tool helped focus on self-management of weight related complications), and *applicability and scalability* (GPs stated it was easy to use, relevant to a range of their patient groups, and scalable if integrated into existing patient management systems).

## Conclusion

The EOSS-2 Risk Tool is potentially clinically useful for activating weight management discussions in general practice. Further research is required to assess feasibility and applicability.

## Introduction

Overweight and obesity is challenging health care systems globally [1], affecting a quarter of all children and adolescents (aged 2–17 years) and around two-thirds of all adults in Australia [2]. It is estimated that approximately 70% of Australians with overweight or obesity (seven million) could have weight related health problems (complex and/or chronic conditions) associated with avoidable health service use [3]. The most common of these include cancer, stroke, heart disease, kidney disease, dementia, diabetes mellitus, back pain, and osteoarthritis [4]. Despite evidence-based guidelines providing recommendations on how to provide effective weight management in primary care [4, 5], excess weight and related complications remain under diagnosed and poorly treated [6–8]. Quality improvements in obesity care are needed and could result in significant population health and economic benefits [9–12].

There is strong worldwide evidence showing that the vast majority of patients with overweight or obesity want their General Practitioner (GP), (also known as 'primary care physician'), to raise the issue of weight during appointments which is often misaligned with inaction [6]. There is also international consensus that obesity stigma is a major barrier to seeking and receiving appropriate treatments for weight management [13]. The most important criteria GPs consider for initiating weight management conversations with a patient are weight related health problems [6]. Other studies show that GPs are more likely to identify obesity or record anthropometric measurements in patients with a weight related health chronic condition [8, 14, 15]. This suggests that targeting weight related health status rather than obesity *per se* may overcome this barrier to initiating treatments in primary care. Thus, a brief diagnostic screening tool for weight related health complications in patients with excess weight could encourage further assessments to confirm a timely diagnosis. It may also help GPs initiate a discussion about the health benefits of weight loss, with or without mentioning obesity, resulting in improvements in the quality of care and health outcomes for their patients.

The Edmonton Obesity Staging System (EOSS) is based on weight related health complications among individuals with overweight and obesity [16]. A score of 2 or higher on the EOSS indicates the presence of clinically significant weight related complications requiring clinical intervention. A rapid review of relevant studies concluded that the EOSS should be routinely

used for predicting risks and benefits of surgical and non-surgical weight management [17]. However, it also highlighted the need for developing standardized tools for clinical settings based on a consistent set of criteria with standardized cut-offs for classifying people into EOSS categories.

To address this need, we developed a new brief screening tool ('EOSS-2 Risk Tool') for weight related complications according to the EOSS scores 2–4 using the Australian Health Survey 2011–13 data set. In the present study, we aimed to assess the feasibility, acceptability, and accuracy of the EOSS-2 Risk Tool for activating weight management discussions in general practice.

## Methods

This manuscript conforms to reporting guidelines for diagnostic accuracy studies [18] and qualitative studies [19].

### Study design and setting

To test the diagnostic accuracy and feasibility of the new EOSS-2 Risk Tool (index test), we used both cross-sectional ('single-gate' [20]) and qualitative study designs across five general practices in Australia (South Australia, New South Wales, Queensland, Victoria, and Western Australia). We planned the data collection before the index test and determined its performance against the reference standard after (retrospectively).

### GP participants

The GP participants were recruited through recruited via the authors professional networks, namely the National Association for Clinical Obesity Services (NACOS) and Healthed, using a promotional flyer seeking expressions of interest "to participate in paid research testing a brief screening tool to help them initiate discussions about obesity with their patients." Practicing GPs willing to comply with the study protocol, including its target recruitment expectation of five patients each over approximately eight weeks, were eligible to enrol in the study. They received payment of $250 (Australian dollars) per patient recruited and completed, to partially compensate them for the extra study tasks over and above standard care.

### Patient participants

The GPs recruited the study patients from their practices. They were asked to identify potentially eligible study participants by selecting patients with suspected overweight or obesity who they believed would benefit from weight management during routine practice and/or through searching their patient database. Additional eligibility criteria were: aged between 18 and 65 years; and willingness and capacity to give written informed consent. Exclusion criteria were: women lactating, current or planned pregnancy during the study; and patients with a history of a psychological illness or condition such as to interfere with the patient's ability to understand the requirements of the study. Patients were not reimbursed for their participation in the study.

### Ethics approval and consent to participate

The study protocol was approved by the Western Sydney University Human Research Ethics Committee (HREC Reference: H14162). Written informed consent was sought and obtained from the GP and patient participants. The GPs were responsible for recruiting patients in the study.

## Index test—EOSS-2 Risk Tool

We developed a brief screening tool ('EOSS-2 Risk Tool') to identify previously undiagnosed weight related complications against published EOSS ≥2 criteria based on clinical information (reference standard) for potential application in general practice [3]. As there is no internationally consistent set of criteria with standardized cut-offs for classifying people into EOSS categories [17], we chose this reference standard which has been validated in an Australian sample of community-based 'high risk' individuals [3]. The tool consists of nine risk factor items including age, self-reported health status (quality of life, disability, bodily pain, and depression or anxiety), and family history (diabetes, hypertension, high sugar in blood/urine, and high cholesterol) relevant to the Australian population with overweight or obesity (S1 Appendix).

## Outcomes

**Diagnostic accuracy.** We validated the diagnostic accuracy of the EOSS-2 Risk Tool (index test) for predicting the presence of EOSS ≥2 criteria (target condition) based on clinical information (reference standard) [3] collected prospectively from the patient participants. The diagnostic accuracy of index test results and clinical information about the reference standard were not available to the GP participants during the study. We used specific thresholds for the EOSS-2 Risk Tool scores to define 'high risk' (<7 points), 'very high risk' (7–24 points), and 'extremely high risk' (≥25 points) of having a diagnosis of clinically significant weight related complications according to EOSS ≥2 criteria.

**Feasibility and applicability.** We assessed the feasibility and applicability of the EOSS-2 Risk Tool in general practice by exploring the views and opinions of the study GPs and their patients using semi-structured interviews.

## Sampling

For the diagnostic accuracy of the index test, we assumed an intraclass correlation of 0.05 and a sample size of 25 patients clustered within five GPs will produce confidence intervals of 0 to 0.229 where the observed proportion is 0.1; and confidence intervals between 0.103 and 0.497 where the observed proportion is 0.3 (PASS Sample Size Software) [21]. This provided us with a general estimation of rates, data errors, and missing data within our quantitative variables.

We used purposive sampling to ensure participants had relevant experience with the phenomenon of interest [22]. Despite the nature of the present pilot study, we anticipated that the small target sample sizes for collecting GP and patient participants' perspectives would be adequate to provide credible and trustworthy preliminary evidence of the feasibility and applicability of the EOSS-2 Risk Tool in general practice. For instance, expert opinions argue that sample size targets for qualitative research have no firm lower bounds [23]. It has been suggested that sample sizes between one and 12 may be most efficient for homogeneous populations and up to 30 for heterogeneous populations [23]. Interviews with patients continued until no new information was revealed and data saturation was reached.

## Data collection

The GP participants scheduled two appointments (no more than two weeks apart) with their patient participants. At the first appointment, they applied the new EOSS-2 Risk Tool and were free to use that point of care and data collection to help them initiate weight management discussions with their patients or not. At the second appointment, they collected any remaining clinical information required by the researchers to establish the presence or absence of

EOSS ≥2 criteria including new or recent (within six months) blood test results, as well as some demographic information (age, gender, country of birth) and anthropometric measurements.

We collected qualitative data from GP participants soon after they had completed all the study tasks in most of their patients enrolled in the study and from patient participants soon after (no more than two weeks apart) their second appointment. To explore GP and patient participants' perspectives of the feasibility and applicability of the tool in general practice, we utilised semi-structured interviews. The interviews included a set of open-ended questions generated prior to the interview to uncover different perspectives (S2 Appendix). One author (JRJ) conducted the interviews after receiving expert training by another author with extensive experience in qualitative interviewing (KP). He used prompt questions to gain a deeper understanding of participants' perspectives or to clarify aspects of their narratives. We sought to complete the interviews within 10 to 15 minutes to minimise study burden. Additionally, we sent the interview questions to some patients who had requested them via a text message prior to their scheduled interview. All interviews were audio-recorded for accurate verbatim transcription.

## Quantitative data analysis

Categorical variables are presented as proportions.

## Qualitative data analysis

The audio-recordings were transcribed verbatim using the online Otter.ai software and imported into Microsoft Word documents for data management. We adopted Braun and Clarke's six phase method of thematic analysis to ensure rigour in the analytic process [24]. The first phase identified by Braun and Clarke is familiarisation with the interview data. This involved immersion in the data by repeatedly listening to the audio-recordings while reading and re-reading the interview transcripts. The second and third phases consisted of identifying patterns and meanings, organising these into initial codes, and then generating broad themes and sub-themes. The fourth phase of analysis involved reviewing the data set to ensure themes are coherent and supported by the data and the fifth phase involved further development and refinement of the themes and sub-themes. Transcripts were independently reviewed and analysed by authors (JRJ, KP, EA) and themes were discussed and further developed until consensus was reached. In the sixth and final phase of analysis, final themes integrated relevant extracts from participants' transcripts with the guiding narrative to authentically convey their experiences.

## Results

Five GPs participated and enrolled 25 patients. One GP recruited one patient only, whereas the other GPs recruited six patients each. Nineteen (76%) patients were aged 45 years or more, five (20%) were male, and 20 (80%) were classified as having obesity. All 25 patients were correctly diagnosed by the EOSS-2 Risk Tool as having clinically significant weight related complications according to the reference standard [3]. The EOSS-2 Risk Tool predicted that 23 (92%) patients were at 'extremely high risk' (≥25 points) and two (8%) were at 'very high risk' (7–24 points) of having clinically significant weight related complications (Table 1). As all patients fell into the same EOSS ≥2 category (significant weight related health conditions) we were unable to further quantify the accuracy of the EOSS-2 Risk Tool.

We present the flow of patient participants through the study and their raw diagnostic results (Fig 1).

**Table 1. Characteristics of the patient sample.**

| Variables | n (%) |
|---|---|
| Age category | |
| • <45 years | 6 (24%) |
| • ≥45 years | 19 (76%) |
| Gender | |
| • Male | 5 (20%) |
| • Female | 20 (80%) |
| Country of birth | |
| • Australia | 21 (84%) |
| • Other | 4 (16%) |
| BMI category (range in kg/m$^2$) | |
| • Normal weight (18.5–24.9) | 1 (4%) |
| • Overweight (25.0–29.9) | 4 (16%) |
| • Obesity class I (30.0–34.9) | 10 (40%) |
| • Obesity class II (35.0–39.9) | 5 (20%) |
| • Obesity class III (≥40) | 5 (20%) |
| EOSS stages (reference standard) | |
| • 0 or 1 | 0 |
| • 2 | 15 (60%) |
| • 3 | 8 (32%) |
| • 4 | 2 (8%) |
| EOSS-2 Risk Tool category (cut-off scores) | |
| • High risk (<7) | 0 |
| • Very high risk (7–24) | 2 (8%) |
| • Extremely high risk (≥25) | 23 (92%) |

Five GPs and 18 patients were interviewed about their experiences of the EOSS-2 Risk Tool. Our analysis of the interview data revealed five themes supporting the feasibility and applicability of the tool in general practice:

1. GP recognition of obesity as a health priority

2. Obesity stigma

3. Patient health literacy

4. Patient motivation for self-management

5. Applicability and scalability

## GP recognition of obesity as a health priority

Most GP participants had a special interest in obesity and an excellent understanding of its importance as a health priority in their communities.

*Most importantly, it's because the community I work with, significant amount of them suffer from obesity related comorbidities. For example, mental health problems, osteoarthritis, diabetes, high blood pressure. All of these kinds of things which, obstructive sleep apnoea related to obesity. Obesity is the common risk factor in this group of patients. So my understanding was that if we could have solved obesity issues, then we could have fixed the other consecutive*

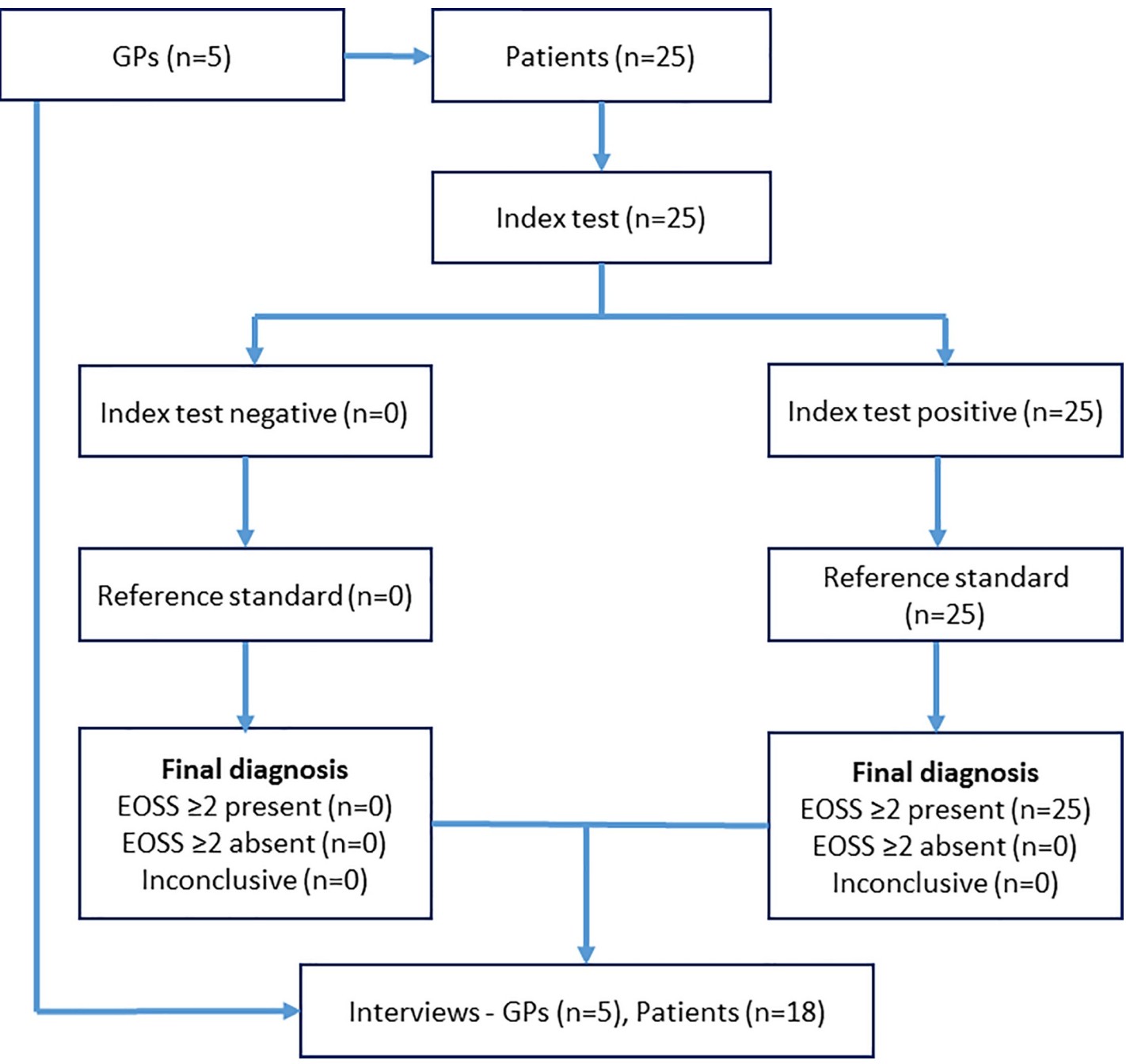

**Fig 1. Flow of patient participants.**

*issues they're having. So that's why over the last two years, I'm trying to you know, get myself more well-resourced, and improve my knowledge on obesity management. So I'm actually interested in anything related to obesity. So I usually try to attend all the webinars and everything I can get. So when I knew that something like this is happening that is related to obesity management, developing a tool where we can categorize the risk associated with obesity, and I thought that is probably one of my interests. (GP4)*

*I actually have quite a lot of overweight patients, I run a weight management program already. And I feel is the highly neglected area of medicine. It is not a simple problem. And it's a very deep rooted problem that takes a lot of time. And people do not present with obesity as the main cause of the medical condition. And they also fight and struggle to raise the issue or address the issue or look to seek active treatment in a preventative way. (GP1)*

## Obesity stigma

Both GP and patient interview participants highlighted several benefits from their experience with the EOSS-2 Risk Tool. Most of the GPs pointed to the usefulness of the tool in discussing weight related issues with their patients, as it helped them initiate health based and non-judgmental unbiased conversations with their patients.

*It really helped to facilitate a discussion around medical illness rather than judgmental values about failure or success as far as their obesity is concerned. It did help to focus the attention away from personal failure and towards medical conditions. (GP2)*

*I think it is a very good conversation starter kind of thing . . . And the questions will all do, I think, primarily related to her genetics, rather than herself. Really good, nothing to do with eating habits or lifestyle choices. And so it was received well, on her part. (GP1)*

*Most of the candidates, they already knew that they're obese, but they were not interested to start the conversation. So at least that helped with them to develop that insight that here's something they need to take seriously. I think if we have got a score, calculated like this, it will be easier to start the conversation because it be very much visible like, you can just show them the test. See you are high risk. So let's talk about this time to change. (GP4)*

Similarly, patient participants described how the tool encouraged them to discuss their weight with their GPs. Discussions on health status rather than weight status was perceived by patients as less confronting.

*I thought it was, I think it was, it was good. It allowed me, it was something that I personally think I have struggled with for a little while. So it was something that I was able to get a better understanding about, which was good. I think it encouraged me to have conversations with [my GP] about ways to better manage weight and ways to try and ensure that it was a sustainable change as opposed to a quick fix. So I definitely encourage those conversations. (Patient 9)*

*So I thought that it was like, if it [EOSS-2] can help a GP to sort of ask the right sort of questions. I thought that would be a good idea or not necessarily ask the right questions, but also have a method of approaching it gently. (Patient 3)*

## Patient health literacy

The GP participants reported that the tool had increased the level of awareness and understanding of weight related health risks among some of their patients.

*It gives an opportunity to see the cardiovascular risk factors . . . So I think that is good. They just said, Oh, I have to think about diabetes, I have to think about stroke, have to think about heart attack and I need to watch for swelling of the legs, where they indicate for heart failure,*

*if you keep on asking the same question you make them more aware about the condition. (GP3)*

Patient participants' accounts concurred with those of the GPs. They acknowledged that because of answering questions from the EOSS-2 Risk Tool and subsequent discussions with their GP, they had increased their knowledge of weight related complications and the importance of physical activity in managing their weight.

*The questions that they asked, I found that they were a little bit interesting, because it highlighted to me some of the areas where I've may have overlooked my own managing of my weight. My knowledge has improved 100%, that a lot of things are linked to it, like having surgeries, that the recovery from the surgery could impact your weight gain, and then walking around with pain all day, that affects your weight gain as well. (Patient 15)*

*And I'm on all those medications, for all these things that like cholesterol and high blood pressure, and all of that, and they're all being controlled through medication, but they're all sort of isolated, but it'd be good to have it all in the one thing, so all parts of these are covered. With all of these things. So it's sort of a little bit of a wake- up call, I suppose. (Patient 3)*

## Patient motivation for self-management

The application of the EOSS-2 Risk Tool motivated patients to focus on self-management their weight related complications.

*I think it helped in the fact that by focusing on the medical complications, and the medical implications of obesity, it allowed the patient to focus on goals that were beneficial for their health, rather than goals that were just behaviourally based. I think from that point of view, it helped as far as motivation goes. (GP2)*

*I think that would be an effective tool in having the patient come back at some stage and saying, you know what, I thought about what you told me, and I believe I need to do something about my weight. And, you know, I'd like to pursue this further. So yeah, I believe it's a good step forward, rather than just going blankly and saying, Oh, yeah, I think you need to lose weight because of this or that. (GP 5)*

Patient participants expressed similar views.

*So it's kind of motivated me take some action. Yeah, I mean, the results of the blood tests kind of made me think about it a bit more, because I hadn't really had too many issues previously, and knowing that not everything's perfect, definitely give me some motivation. I know that if I follow the plan, it will definitely help me be more active and healthier and longer for your long term health issues. (Patient 6)*

*Maybe a motivating factor would be that now there are developed cholesterol issues, and I'm more inclined to think about trying how to manage that. And so I've looked up high cholesterol, and foods that increase your high cholesterol and foods that help to decrease it. So that sort of thing that you know, you just become more self- aware and do more of your own research. (Patient 8)*

*Well, I think it's given me confidence that I am doing the right thing. Because of all the ratings, I got it, you know, and even [my GP] said, You're very enthused to get your health in check.*

*And so it certainly helped me with that. If I've not done this, I would have just been on my own really after been with [my GP] thinking well, am I doing the right thing, am I not but by going back to see him and him you know, doing all my, you know, blood pressure and everything again and doing my blood. It's given me confidence that yes, I am on the right track to improving my health and that's very important to me at my age. (Patient 16)*

*The fact that we're just discussing plans and reminding me of that, yes. I've got the motivation to do it and the plans that we have required to, to lose weight. So I think this is more reinforcement. I'm more confident that I can lose weight. She's [GP] there to help and refer me to the people that can help me as well. (Patient 7)*

## Applicability and scalability

There was consensus among most GPs that the EOSS-2 Risk Tool was easy to use, relevant to a range of their patient groups, and possibly scalable in general practice.

*Oh yes I was more than comfortable dealing with it [EOSS-2 Risk Tool]. There was nothing too stressful, there was nothing too difficult in the question asking, and as I say, most of that data would be in my database anyway. I would know most of that anyway, in my patients. And, you know, again, it would but then it would be very easy for me to say, look, Mom, dad had diabetes too, you're at an extreme high risk straightaway without even asking the questions. Yeah. So in one sense, I felt comfortable with it. (GP2)*

*Yeah, that [discussion about weight] was easy to elicit, because during the data collection, you ask all these questions about health, and usually the patient goes like, oh, why all these questions there? And I said because if you have elevated weight, it could affect all these things in your body. You can have any of these problems. (GP5)*

Patient participants expressed agreement with the GPs regarding the simplicity of the EOSS-2 Risk Tool stating that it *was actually pretty easy (Patient 4), The questions weren't difficult to answer (Patient 1)* and that *It was fine, the questions were all really efficient, and quick and not a problem at all. (Patient 6).*

Some GPs suggested that the tool would be valuable in the early screening and prevention of weight related health risks in younger people who would not necessarily be considered a high risk population group.

*For people who have got not many risk factors, I'm talking about the younger population, relatively younger population, they probably don't know how this obesity problem is going to affect them in next five to 10 years, 15 years, 20 years' time, which can be a considerable health burden. So for this group of people, I think they're from 20 years of age up to 50 or 45, this group of people is really effective, because we can at least start or initiate the discussion with them. (GP4)*

Most GP participants suggested that the EOSS-2 Risk Tool could be broadly implemented in general practice if it were integrated into existing patient management systems.

*Yes. CVD risk calculator, diabetes risk calculator, we only use this and K10 score and this kind of thing. We use it a general screening. It just [needs to] become more coincided. And to make it more organized if you have got something visible. So yeah, so why not [use the EOSS-2 Risk Tool]? I think if we get a good score calculator system for obesity, we would be using it*

*adopting it regularly. If it becomes a part of the you know, the software, then it becomes easier. (GP4)*

*I think I would consider using it [long term]. Definitely. If it's attached to something to say what EOSS-2 or above means and listing the elements, the risk factors, and the diseases that could potentially happen in people with EOSS-2 and above, then I think that would be an effective tool. (GP 5)*

## Discussion

The AWARE pilot is the first study to evaluate the feasibility, applicability, and accuracy of the new EOSS-2 Risk Tool for activating weight management discussions in general practice. Although the EOSS-2 Risk Tool was primarily developed for initiating such discussions, all patients screened positive for meeting diagnostic criteria for EOSS ≥2, as was expected for such a 'high risk' patient population group purposely targeted by their GPs. The GPs correctly selected all but one of their patients with overweight or obesity and all had clinically significant complications. The ability of the GPs to correctly select high risk patients was likely heightened by the study protocol and recruitment strategy, as well as the tool's focus on weight related complications. This hypothesis is supported by evidence showing that GPs are more likely to identify or record obesity in patients with a weight related health chronic condition, than those without [8, 14, 15]. Thus, having the EOSS-2 Risk Tool available for routine use in general practice might help GPs proactively target high risk patients for weight assessment and management with more confidence.

We found evidence that the EOSS-2 Risk Tool helped addressed obesity stigma in general practice. The GP participants highlighted the tool's usefulness in initiating discussions about weight related health issues with their patients in a comfortable and non-judgemental way. Similarly, patient participants described how the tool made them feel more comfortable discussing their weight with their GPs. It helped GPs focus on medical goals instead of their patients' behaviour and clearly removed uncertainty about using appropriate language and concern about bringing up their weight, which is commonly expressed by health care professionals globally [6, 25]. Our evidence supports the hypothesis that focusing on health status rather than weight status may help GPs and their patients overcome this barrier to initiating treatments in general practice. In fact, 'obesity-related comorbidities' was one of the principal reasons cited by health care professionals in the Awareness, Care and Treatment In Obesity maNagement-International Observation (ACTION-IO) study for initiating weight management discussions [26]. Thus, the EOSS-2 Risk Tool could potentially reduce the average delay of 8.9 years reported by Australians in initiating weight management discussions with their GPs from the time when they first had concerns about their weight [26].

We found evidence that the tool improved patients' health literacy about the health risks associated with excess weight and motivation for self-management to proactively monitor and manage their weight related complications and lifestyle behaviours. Low health literacy is consistently associated with poor health outcomes and increased use of health care services [27], which should be targeted for intervention. There is systematic review evidence suggesting that health literacy interventions are effective in supporting positive changes in behavioural risk factors [28]. Therefore, the application of the EOSS-2 Risk Tool could help patients adhere to evidence-based guidelines for the management of overweight and obesity in primary care, which recommend multifactorial lifestyle interventions to support healthy changes in behaviour [4, 5].

There was general agreement among all GPs that the EOSS-2 Risk Tool was applicable to a range of patients, including young adults for early detection and prevention of developing weight related complications. It could be a valuable addition to current screening tools for preventing type 2 diabetes mellitus and cardiovascular disease, albeit variably used in Australian general practice [29–31]. Although the effectiveness of financial incentives to improve quality of care remains controversial [32], they may be required to encourage its application in general practice [33]. For instance, a recent found that the majority of GPs surveyed (78%) reported using the absolute cardiovascular disease risk tool, which is subsidized by Australia's public health care system (Medicare), in their practice [34]. We identified that implementation of the tool in routine clinical practice may also require electronic integration into existing patient management systems. This finding is consistent with recommendations for the integration of an electronic version of the current cardiovascular disease risk tool in Australian general practice [30, 35].

### Strengths and limitations

Study limitations should be considered in the interpretation of our supportive findings above. As this study was a small pilot in a select group of incentivised GPs and motivated patients, definitive evidence of effectiveness of the EOSS-2 Risk Tool for initiating weight management discussions and treatments compared with standard care in general practice remains unclear. Although the EOSS-2 Risk Tool was developed to help GPs initiate conversations about weight management in 'high risk' patients, its accuracy for detecting clinically significant weight related complications in other general ('lower risk') patients is unclear. The interviews were conducted by one author (JRJ) who was aware of the study aims which might have influenced responses from the participants.

### Conclusions

The EOSS-2 Risk Tool was found to be clinically useful for activating weight management discussions in general practice. Research to generate definitive evidence of effectiveness, feasibility, and applicability may lead to scalable and sustainable improvements in the standards of care for excess weight and related complications in Australia's health care system.

### Supporting information

**S1 Appendix. EOSS-2 risk screening tool.**
(DOCX)

**S2 Appendix. Interview schedules.**
(DOCX)

### Acknowledgments

We acknowledge Healthed for promoting the study among potentially interested GPs through their vast national electronic database, the GP participants for their efforts in recruiting their patients, and their patients for their time and participation in the study.

### Author Contributions

**Conceptualization:** Evan Atlantis.

**Data curation:** Evan Atlantis, James Rufus John, Kath Peters.

**Formal analysis:** Paul Patrick Fahey, Kath Peters.

**Funding acquisition:** Evan Atlantis, Paul Patrick Fahey, Kath Peters.

**Investigation:** Evan Atlantis, Paul Patrick Fahey, Kath Peters.

**Methodology:** Evan Atlantis, Kath Peters.

**Project administration:** James Rufus John.

**Supervision:** Evan Atlantis, Kath Peters.

**Validation:** Evan Atlantis, Paul Patrick Fahey, Kath Peters.

**Writing – original draft:** Evan Atlantis.

**Writing – review & editing:** Evan Atlantis, Paul Patrick Fahey, Samantha Hocking, Kath Peters.

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
