## [Decision Letter · Decision Letter 0]

22 Sep 2021

PONE-D-21-24511Clinical usefulness of brief screening tool for Activating Weight management discussions in primary cARE (AWARE): a nationwide mixed methods pilot studyPLOS ONE

Dear Dr. Atlantis,

Thank you for submitting your manuscript to PLOS ONE. After careful consideration, we feel that it has merit but does not fully meet PLOS ONE’s publication criteria as it currently stands. Therefore, we invite you to submit a revised version of the manuscript that addresses the points raised during the review process.Besides comments with respect to content of the manuscript made by the reviewer, but also e thoroughly check the manuscript for grammatical errors and non-academic language.

We look forward to receiving your revised manuscript.

Kind regards,

Sabine Rohrmann

Academic Editor

PLOS ONE

Journal Requirements:

2. When reporting the results of qualitative research, we suggest consulting the COREQ guidelines  or other relevant checklists listed by the Equator Network, such as the SRQR, to ensure complete reporting (http://journals.plos.org/plosone/s/submission-guidelines#loc-qualitative-research)." Do not ping with follow up, thanks!

Reviewers' comments:

Reviewer's Responses to Questions

**Comments to the Author**

1. Is the manuscript technically sound, and do the data support the conclusions?

Reviewer #1: Partly

2. Has the statistical analysis been performed appropriately and rigorously? 

Reviewer #1: Yes

3. Have the authors made all data underlying the findings in their manuscript fully available?

Reviewer #1: Yes

4. Is the manuscript presented in an intelligible fashion and written in standard English?

Reviewer #1: No

5. Review Comments to the Author

Reviewer #1: Thankyou to the authors for the opportunity to review this manuscript. This study addresses a very important gap in knowledge and practice. The inclusion of both qualitative and quantitative data was great. I have made a number of suggestions for the authors to consider:

Abstract:

Please define EOSS prior to introducing EOSS-2 risk tool, and explain how it is derived. You have this information in the introduction (page 4 lines 84-87).

Please consider rewriting lines 40-41 for clarity.

Please consider 'Further research is required to assess feasibility and applicability'.

Introduction:

Are 2-17 years olds better described as children rather than young people?

Do you need to provide contextual information re GPs, given the international audience?

Methods:

In general, I think the methods section could benefit from refining to reduce repetition, length and improve clarity.

Please consider 'recruited via authors' professional networks, namely...' page 5, line 111

Were patients reimbursed for their time?

Why was a patient living without overweight/obesity included in the study, presumably with weight related health concerns?

Were GPS responsible for obtaining consent?

Page 6, line 137, please consider 'to identify previously undiagnosed weight related complications'

Page 7, line 144, is family history the most accurate term, or is this point about previously undiagnosed health concerns?

Page 8, lines 179-182, please consider omitting

Page 8 lines 182-184, these are results

Page 8, line 187, page 9, line 197, could you say 'no more than 2 weeks apart'?

Page 9, line 211, likely fits better in results

Results:

Could you state the % of participants who were categorised as EOSS 2,3 or 4 separately?

Page 13, line 283: I would recommend removing the numerical assessment of the qualitative findings.

I would argue that the quotes from patient 9 and 3 talk more about weight loss than health (page 14, lines 305-314).

However, the quote from patient 6 on page 16 is a better example

Discussion:

Could the authors please clarify how their results address questions of accuracy? This needs to be addressed, particularly as the authors have identified this as a limitation later in the discussion.

Page 20, line 471, I think that stating consensus with a sample of 5 GPs is over reaching.

The manuscript needs to be thoroughly checked for grammatical errors and non-academic language.

6. PLOS authors have the option to publish the peer review history of their article (what does this mean?). If published, this will include your full peer review and any attached files.

Reviewer #1: No

---

## [Author Response · Author response to Decision Letter 0]

8 Oct 2021

Response to Reviewers

Journal Requirements:

To our knowledge, the manuscript meets PLOS ONE's style requirements.

2. When reporting the results of qualitative research, we suggest consulting the COREQ guidelines or other

Please see page 5:

This manuscript conforms to reporting guidelines for diagnostic accuracy studies (19) and qualitative studies (20).

20. Tong A, Sainsbury P, Craig J. Consolidated criteria for reporting qualitative research (COREQ): a 32-item checklist for interviews and focus groups. Int J Qual Health Care. 2007;19(6):349-57.

Review Comments to the Author

Reviewer #1: Thank you to the authors for the opportunity to review this manuscript. This study addresses a very important gap in knowledge and practice. The inclusion of both qualitative and quantitative data was great. I have made a number of suggestions for the authors to consider:

Abstract:

Please define EOSS prior to introducing EOSS-2 risk tool, and explain how it is derived. You have this information in the introduction (page 4 lines 84-87).

We appreciate this suggestion and have revised the abstract accordingly.

Please consider rewriting lines 40-41 for clarity.

We have revised this sentence for clarity.

Please consider 'Further research is required to assess feasibility and applicability'.

We appreciate the suggestion and have revised this sentence accordingly.

Introduction:

Are 2-17 years olds better described as children rather than young people?

We have revised this to read ‘children and adolescents’.

Do you need to provide contextual information re GPs, given the international audience?

We have added ‘worldwide’ and ‘primary care physician’ to provide the international context of the evidence.

Methods:

In general, I think the methods section could benefit from refining to reduce repetition, length and improve clarity.

Please consider 'recruited via authors' professional networks, namely...' page 5, line 111

We have revised this section accordingly.

Were patients reimbursed for their time?

We have clarified that patients were not reimbursed for their participation in the study.

Why was a patient living without overweight/obesity included in the study, presumably with weight related health concerns?

We presume so as the GP’s were responsible for ‘selecting patients with suspected overweight or obesity who they believed would benefit from weight management during routine practice and/or through searching their patient database’. 

Were GPS responsible for obtaining consent?

The selected patient participants received study information packs from their GP containing specific participant information and a consent form. The study GPs were asked to reiterate to the patient that study participation was entirely voluntary, and that any decision to participate or not would not harm their existing doctor-patient relationship. Due to travel restrictions and locality of study sites, participants were required to return the signed consent form via a reply-paid postage to the University research team. The research team then sent scanned copies of the signed consent forms to the study GPs by email advising them that they could schedule the first study visit. 

Given the above stated concern about the length of the method section, no changes were made. 

Page 6, line 137, please consider 'to identify previously undiagnosed weight related complications'

We have revised this accordingly.

Page 7, line 144, is family history the most accurate term, or is this point about previously undiagnosed health concerns?

We retained the term used in the survey to develop the new tool and believe it is a proxy variable of genetic risk factors. 

Page 8, lines 179-182, please consider omitting

We have omitted this sentence.

Page 8 lines 182-184, these are results

We agree and have deleted this sentence for being redundant.

Page 8, line 187, page 9, line 197, could you say 'no more than 2 weeks apart'?

We have revised these sections accordingly.

Page 9, line 211, likely fits better in results

We agree and have revised this accordingly.

Results:

Could you state the % of participants who were categorised as EOSS 2,3 or 4 separately?

We have revised the table presenting proportions for EOSS 2, 3, and 4.

Page 13, line 283: I would recommend removing the numerical assessment of the qualitative findings.

We have revised this accordingly.

I would argue that the quotes from patient 9 and 3 talk more about weight loss than health (page 14, lines 305-314).

However, the quote from patient 6 on page 16 is a better example

After carefully considering the reviewer’s interpretation, we reviewed this section again. While we appreciate the suggestion, we still believe that the selected quotes appropriately support the themes. 

Discussion:

Could the authors please clarify how their results address questions of accuracy? This needs to be addressed, particularly as the authors have identified this as a limitation later in the discussion.

We have revised this for clarity. 

Page 20, line 471, I think that stating consensus with a sample of 5 GPs is over reaching.

We have moderated this sentence.

---

## [Decision Letter · Decision Letter 1]

18 Oct 2021

Clinical usefulness of brief screening tool for Activating Weight management discussions in primary cARE (AWARE): a nationwide mixed methods pilot study

PONE-D-21-24511R1

Dear Dr. Atlantis,

We’re pleased to inform you that your manuscript has been judged scientifically suitable for publication and will be formally accepted for publication once it meets all outstanding technical requirements.

Kind regards,

Sabine Rohrmann

Academic Editor

PLOS ONE

Additional Editor Comments (optional):

Reviewers' comments:

Reviewer's Responses to Questions

**Comments to the Author**

1. If the authors have adequately addressed your comments raised in a previous round of review and you feel that this manuscript is now acceptable for publication, you may indicate that here to bypass the “Comments to the Author” section, enter your conflict of interest statement in the “Confidential to Editor” section, and submit your "Accept" recommendation.

Reviewer #1: All comments have been addressed

2. Is the manuscript technically sound, and do the data support the conclusions?

Reviewer #1: Yes

3. Has the statistical analysis been performed appropriately and rigorously? 

Reviewer #1: Yes

4. Have the authors made all data underlying the findings in their manuscript fully available?

Reviewer #1: No

5. Is the manuscript presented in an intelligible fashion and written in standard English?

Reviewer #1: Yes

6. Review Comments to the Author

Reviewer #1: Thankyou to the authors for their careful consideration of the suggestions. I wish the authors well for their research into the future

7. PLOS authors have the option to publish the peer review history of their article (what does this mean?). If published, this will include your full peer review and any attached files.

Reviewer #1: No

---

## [Editor Report · Acceptance letter]

20 Oct 2021

PONE-D-21-24511R1 

Clinical usefulness of brief screening tool for Activating Weight management discussions in primary cARE (AWARE): a nationwide mixed methods pilot study 

Dear Dr. Atlantis:

I'm pleased to inform you that your manuscript has been deemed suitable for publication in PLOS ONE. Congratulations! Your manuscript is now with our production department. 

Kind regards, 

on behalf of

Dr. Sabine Rohrmann 

Academic Editor

PLOS ONE